# Evaluation of Electrochemical Stability of Sulfonated Anthraquinone-Based Acidic Electrolyte for Redox Flow Battery Application

**DOI:** 10.3390/molecules26092484

**Published:** 2021-04-24

**Authors:** Petr Mazúr, Jiří Charvát, Jindřich Mrlík, Jaromír Pocedič, Jiří Akrman, Lubomír Kubáč, Barbora Řeháková, Juraj Kosek

**Affiliations:** 1Department of Chemical Engineering, University of Chemistry and Technology, Technická 5, Praha 6, 166 28 Prague, Czech Republic; charvatj@vscht.cz (J.C.); mrlikj@vscht.cz (J.M.); jkk@vscht.cz (J.K.); 2New Technologies—Research Centre, University of West Bohemia, Univerzitní 8, 306 14 Plzeň, Czech Republic; pocedicj@ntc.zcu.cz; 3Centre for Organic Chemistry, Rybitvi 296, 533 54 Rybitvi, Czech Republic; jiri.akrman@cocltd.cz (J.A.); lubomir.kubac@cocltd.cz (L.K.); 4Synthesia, a.s., Semtín 103, 530 02 Pardubice, Czech Republic; Barbora.Rehakova@synthesia.cz

**Keywords:** redox flow battery, aqueous organic electrolyte, anthraquinone disulfonic acid, capacity decay, electrolyte cross-over

## Abstract

Despite intense research in the field of aqueous organic redox flow batteries, low molecular stability of electroactive compounds limits further commercialization. Additionally, currently used methods typically cannot differentiate between individual capacity fade mechanisms, such as degradation of electroactive compound and its cross-over through the membrane. We present a more complex method for in situ evaluation of (electro)chemical stability of electrolytes using a flow electrolyser and a double half-cell including permeation measurements of electrolyte cross-over through a membrane by a UV–VIS spectrometer. The method is employed to study (electro)chemical stability of acidic negolyte based on an anthraquinone sulfonation mixture containing mainly 2,6- and 2,7-anthraquinone disulfonic acid isomers, which can be directly used as an RFB negolyte. The effect of electrolyte state of charge (SoC), current load and operating temperature on electrolyte stability is tested. The results show enhanced capacity decay for fully charged electrolyte (0.9 and 2.45% per day at 20 °C and 40 °C, respectively) while very good stability is observed at 50% SoC and lower, even at 40 °C and under current load (0.02% per day). HPLC analysis conformed deep degradation of AQ derivatives connected with the loss of aromaticity. The developed method can be adopted for stability evaluation of electrolytes of various organic and inorganic RFB chemistries.

## 1. Introduction

A growing share of renewable energy sources on the total electricity production can significantly affect the stability and security of the distribution grid. The so-called redox flow battery (RFB) is one of the emerging technologies to overcome the growing problems associated with the intermittency of renewable energy sources. In contrast to standard accumulators, power and capacity of RFB can be designed independently, which is beneficial, especially for large-scale stationary energy storage applications. When compared to mostly currently employed Li-ion batteries, RFB are typically safer (when using non-flammable aqueous electrolytes) and potentially more durable as charge storage, and release is only accompanied by redox transition of electroactive specie on inert electrodes. Among several metal-based chemistries, the all-vanadium RFB currently represents the most matured and commercially available technology [1], however, the high price of vanadium and some of its adverse features (e.g., toxicity, low solubility, or slow electrode reaction kinetics) motivates seeking alternative electroactive substances, including organics [2].

Introduced by Aziz’s research group from Harvard University, anthraquinone derivatives were among the first organic electroactive compounds reported for utilization in negolyte of aqueous RFBs. Hydroxylated and carboxylated derivatives of anthraquinone (AQ) showed promising properties in alkaline environment [3,4], while 9,10-anthraquinone-2,7-disulphonic acid (2,7-AQDS) was successfully combined with bromine/bromide posilyte in acidic environment, providing high solubility (over 1 M for 2 electron transition), relatively low reduction potential (+0.2 V vs. SHE at pH 0), fast electron kinetics (k0 = 7.2 10^−3^ cm s^−1^ on glassy carbon) and potential of low cost ($21 per kWh) [5,6,7]. Although other AQ derivatives have also been reported for the application [8], 2,7-AQDS is the most frequently quinone-based compound reported for negolyte both in acidic and neutral environments in combination with various posilytes based on bromine [5,9], iodine [10], iron [11,12], or benzoquinone [13]).

The stability of anthraquinone-based electrolytes has been recently addressed in several publications. Tong et al. [14] used UV-VIS spectroscopy to study the mechanism of formation of quinhydrone, i.e., a dimer formed via hydrogen bonding between oxidized and reduced quinone forms in the solution. Using bulk electrolysis in a stirred half-cell Carney et al. [15] observed that an intermolecular dimerization of AQDS in acidic solution takes place at concentrations as low as 10 mmol dm^−3^ leading to decreased capacity utilization corresponding to 1.5 mol e^–^ per mol AQDS. Recently, the formation of an electrochemically inactive dimer of the oxidized AQDS molecules was observed by Wiberg et al. [16] significantly reducing the electrolyte utilization. According to Wermeckes et al. [17] the degradation of unsubstituted AQ in an acidic environment consists of disproportionation of its 2e^–^ reduced form (anthrahydroquinone, AQDSH_2_) to anthraquinone and electrochemically inactive anthrone. Anthrone formation, which is presumed to proceed also for substituted anthraquinone derivatives, has been observed in case of 2,6-dihydroxyanthraquinone (2,6-DHAQ) in an alkaline environment (pH 14) by Goulet et al. [18] significantly contributing to the capacity fade of RFB using 2,6-DHAQ negolyte and ferrocyanide-based posilyte. A partial capacity recovery was achieved after oxidation of anthrone by purging the anolyte by oxygen, however, full restoration of the capacity was disabled by subsequent anthrone dimerization. The same authors [19] introduced a method for determination of electrolyte stability using a symmetric flow battery cell, i.e., using the same electrolyte in both half-cell, which partially eliminates the contribution of permeation of electroactive compound through ion-exchange membrane on the observed capacity fade. Combination of potentiostatic cycling with OCV pauses in different state of charge (SoC) was used to study chemical and electrochemical stability of various redox couples, including acidic 2,7-AQDS electrolyte. In general, a very slow capacity fade (below 0.1%/day) was observed together with an interesting transient behavior of the capacity fade after hold in 0 and 100% SoC, which the authors attributed to the crossover phenomena due to different permeability of the oxidized and reduced forms. According to the assumptions, a slightly faster capacity fade was observed when the battery was paused at fully reduced form (100% SoC) suggesting faster chemical degradation of AQDSH_2_. This was also supported by the presence of new NMR signals in the negolyte after the experiment. However, the degradation mechanism and nature of degradation products have not been satisfactorily clarified.

In our study, we introduce a complex methodology for evaluation of capacity and performance fade in redox flow batteries which was used to investigate the chemical and electrochemical stability of AQDS-based acidic electrolyte as a model representative of organic aqueous RFB electrolyte. When compared to previous reports, our approach systematically and in detail studies the effect of the operation conditions (SoC, current load and temperature) on the stability of AQDS-based negolyte using an innovative combination of flow electrolysis cell (FEC) and double half-cell (DHC) flow battery. The individual sources of capacity fade are assessed including permeation of active compounds across ion-exchange membrane using UV–VIS spectroscopy. The post mortem analysis of the electrolytes by UV–VIS spectroscopy and high-performance liquid chromatography was performed to identify the changes in electrolyte composition. Moreover, we show that mixture from AQ sulfonation can be directly used in acidic AQDS-Br RFB, providing comparable performance as negolyte prepared from pure 2,7-AQDS compound.

## 2. Results and Discussion

### 2.1. AQDS—Br Single-Cell Performance

Firstly, the electrochemical performance of AQDS-based electrolyte directly prepared from AQ sulfonation mixture was tested in AQDS-HBr single-cell using 3.0 M HBr positive electrolyte according to [5,7,8] and its parameters were compared with 2,7-AQDS-based electrolyte to see whether the sulfonation mixture can be directly used without time and energy demanding separation of 2,7-AQDS isomer. The experimental procedure consisted of EIS and load curve characterization of the single-cell in 50% SoC followed by a series of 50 galvanostatic cycles at 125 mA cm^–2^ in corresponding voltage limits. The performance of both cells are mutually compared in Figure 1 and Table 1. The cell containing AQ sulfonation mixture exhibited slightly higher over-all polarization (*R*_dis_ of 0.58 Ω cm^2^ vs. 0.44 Ω cm^2^ for 2,7-AQDS based negolyte) due to almost double *R*_CT_ (0.38 Ω cm^2^ vs. 0.18 Ω cm^2^). This is probably related to the presence of 2,6-AQDS in the anolyte potentially exhibiting slower reaction kinetics. *R*_ohmic_ of both batteries was around 0.2 Ω cm^2^. The higher over-all resistance of the battery based on AQ sulfonation mixture resulted in approx. 6% lower *EE* at 125 mA cm^−2^ (77.2% vs. 83.2%). On the other hand, this battery provided comparable initial *Q*_dis_ (approx. 445 mAh) and even lower *CD* (–0.34% *Q_t_*_heor_/cycle for mixed electrolyte vs. –0.45% *Q_t_*_heor_/cycle for 2,7-AQDS based one), which might be related to slower permeation of the mixed electrolyte active compounds through the used PFSA membranes, although in the previous studies, the bromine permeation across the cation-exchange membrane has been identified as the main cause of capacity fade in this system [6,20,21]. In sum, the obtained results proved that the sulfonation AQ mixture can be directly used as acidic electrolyte which can significantly reduce the production costs (approx. to a half or even a third of the standard cost) as the costly separation of 2,6-AQDS and other isomers can be omitted.

### 2.2. Electrochemical Stability of AQDS Sulfonation Mixture

In order to exclude the effect of catholyte (e.g., bromine permeation) on capacity decay we have proposed and developed a method for stability evaluation based on a combination of flow electrolysis cell (FEC) and double half-cell (DHC), see Figure 2. FEC is used for adjustment of SoC of the working AQDS-based electrolyte and its charge-discharge cycling, while DHC is used to apply current load without altering SoC of working electrolyte and to see the effect of operating parameters on DHC performance. Let us start with the results obtained with FEC, where the characterization experiment consisting of 10 reduction-oxidation cycles was performed to assess the actual capacity. Broad potential limits (from –0.1 V to 0.65 V vs. SHE) and combined galvanostatic-potentiostatic regime (*I*-*U* cycling) were used to maximize the utilization of the electroactive compound within the cycling without the disturbing effect of the half-cell polarization as suggested by Goulet et al. [19]

The evolution of working electrode potential during combined *I-U* cycling is for individual cycling series presented in Figure 3a, the evolution of the FEC parameters is shown in Figure 3b. In the following text we will try to describe the complex experimental procedure and discuss the results. Within initial cycling, we observed relatively high and stable values of efficiencies and *Q*_dis_. High relative *CU* over 95% shows that both main electrolyte constituents (2,6-AQDS and 2,7-AQDS) are electrochemically active within the potential window used for the cycling which is in good agreement with previous report [5]. At the same time our results do not support the hypothesis of Carney et al. [15] who observed the reduced *CU* due to dimerization of AQDS. The averaged *CE* of initial cycling series is 99.7% which implies that only 0.3% of the reduction charge (approx. 15 mAh/cycle) is not recovered during back oxidation. This inefficiency can be potentially caused by several mechanisms including: (i) Degradation of electro-active compounds of AQDS electrolyte; (ii) permeation of electro-active compounds of AQDS electrolyte through the membrane to counter the electrolyte; and/or (iii) parasitic hydrogen evolution (which is thermodynamically possible at higher SoC, where electrode potential decreases below 0 V vs. SHE). The leakage of electrolytes out of the system was not visually detected within the entire study. During the first cycling we observed capacity decay with a slope of 3.8 mAh/cycle which corresponds to 0.08% of *Q*_theor._/cycle (0.7% of *Q*_theor._/day).

After the first cycling, the AQDS electrolyte was kept fully discharged (0% SoC) and it was circulated via apparatus for five days under OCV conditions (zero current load on DHC and FEC) to assess the chemical stability of AQDS in fully oxidized form. UV–VIS absorbance of counter electrolyte was measured prior and after the hold to evaluate the permeation of oxidized AQDS through the membrane of FEC. According to expectations, AQDS concentration in counter electrolyte increased only negligibly from initial 0 to 1.8 10^–5^ mol dm^−3^, which corresponds to a negligible loss of the available capacity of the working electrolyte of 0.19 mAh (0.004% of *Q*_theor._ in five days). Afterwards, *U-I* cycling in FEC was repeated to see the effect of the hold on *Q*_dis_ and other operating parameters of FEC working half-cell. The resulting parameters of FEC are for all series presented in Figure 3b and summarized in Table 2. After five days at 0% SoC under OCV conditions, *Q*_dis_ even slightly increased from 4402 to 4410 mAh corresponding to 0.17% capacity increase with respect to *Q*_theor._, which proves excellent stability of oxidized AQDS and its negligible permeation through the selected cation-exchange membrane. Additionally, the other parameters of FEC were fully comparable to the initial cycling, with only a slight decrease of *VE*.

Subsequently, the AQDS-based electrolyte was fully reduced to 100% SoC and the same procedure was repeated. After five days at fully reduced state, the cycling in FEC showed decreased *Q*_dis_ by 202 mAh corresponding to a loss of 4.4% of *Q*_theor_. (0.9% *Q*_theor_./day) while AQDS loss due to permeation through the membrane accounted only for 0.08 mAh (0.004% *Q*_theor._ in five days), in fact, the permeation was even slower than for 0% SoC. This indicates that the chemical instability of reduced AQDS is the main source of capacity decay. The decreased capacity also negatively influenced *VE* of the working half-cell of FEC by 5% probably due to decreased concentration of active species in the working electrolyte resulting in the increased activation and mass transport polarization of the cell.

The same procedure was subsequently repeated in 50% SoC where the five days held under OCV conditions led to a very small capacity decay of 0.2% of *Q*_theor_ in five days (0.04% of *Q*_theor_/day) and the available capacity also remained unchanged when current load of 150 mA cm^−2^ was applied on the DHC. These results clearly show that the degradation of AQDS is purely chemical process, which is not accelerated by the electrochemical reduction and oxidation reaction taking place at DHC electrodes. In both cases, the capacity losses due to the permeation of AQDS across the membrane in FEC accounted for only 0.002% of Q_theor_. The last experiment was afterwards repeated at an increased operating temperature of 40 °C leading to only slightly increased capacity decay of 0.4% of *Q*_theor_ (0.08% of *Q*_theor_/day). According to the expectations, the increased temperature also accelerated cross-over of AQDS, which accounted for capacity loss of 0.02% of *Q*_theor._ (i.e., 10 times more than at 20 °C). Finally, the electrolyte was held in a fully reduced state (100% So*C*), at OCV and 40 °C, resulting in the highest capacity decay of 565 mAh corresponding to 12% of *Q*_theor._ in five days (2.45% of *Q*_theor_/day) due to accelerated chemical degradation of AQDS at elevated temperature. The permeation of AQDS through the membrane corresponded to 0.023% of *Q*_theor_ which is comparable to the situation at 50% SoC at this temperature. The decreased concentration of active species due to AQDS degradation resulted in further decrease of *VE* of electrolysis half-cell by almost 5%.

### 2.3. Chemical Degradation Tests

The obtained results suggested us that the capacity fade is mostly related to chemical degradation of AQDS which is significantly enhanced in its reduced form (anthrahydroquinone) and is accelerated by increased temperature. After the electrochemical test, the electrolyte (converted to fully oxidized state) was analyzed by UV–VIS spectroscopy and its composition was compared with the fresh one, see Appendix A. Both samples provided us with qualitatively similar spectra with three absorption maxima at 328 nm, 258 nm, 210 nm, and a wave at 278 nm. The absorbance of the main peak at 258 nm decreased by 33% showing almost double decreased content of the active compounds than the corresponding capacity decay observed in FEC. This suggests that chemical degradation took place also between the electrochemical tests and its analysis. The distribution of individual compounds in the mixture after the test (represented by area % from HPLC) remained similar when compared to the initial electrolyte. The content of the main constituents 2,6-AQDS and 2,7-AQDS slightly increased while the content of the third main constituent, anthraquinone-2-sulfonic acid, decreased to the half of the initial value. Besides, no new signals corresponding to degradation products were found by HPLC (see Appendix A), which indicate deep degradation of the active compounds leading to loss of the aromaticity. In previous studies, anthrone formation due to the disproportionation of anthrahydroquinone was reported as the main degradation mechanism in AQ-based negolytes [17,18]. We have not identified anthrone in the sulfonation mixture after the tests, which can be explained by the fact that the electrolyte samples were always analyzed in the oxidized form and after dilution by water containing dissolved oxygen, which would oxidize anthrone, if presented. The oxidation of anthrone to anthraquinone by purging the electrolyte with oxygen has been reported for dihydroxyanthraquinone alkaline anolyte leading to partial restoration of the battery capacity [18].

Comparable degradation rates for the main AQDS isomers of the electrolyte made us think of alternative explanations for the loss of AQDS from the electrolyte, such as its adsorption on the surface of cell components (felts, membrane, distribution frames) or tubing used for electrolyte circulation. Alternatively, the decomposition of AQDS in the working electrolyte might be enhanced by peroxysulfuric acid which can be potentially formed on the counter electrode during the charging of FEC, when oxygen evolution reaction takes place at high potentials (above 2 V vs. SHE). The highly oxidative persulfate ion can potentially permeate through the cation exchange membrane of FEC and enhance the decomposition of AQ derivatives, which would explain similarly decreased content for all AQ derivatives.

In order to exclude these possible contributions to the observed decrease content of AQDS, chemical stability of AQDS electrolyte was further studied on a series of AQDS electrolyte samples of various SoC stored under argon blanket in sealed glass containers at different temperatures (20 or 60 °C). The SoC of the samples was adjusted by its charging for appropriate time in a std. RFB single-cell using HBr/Br_2_ posilyte to simulate the real conditions of the applications. After the five days the composition of the samples was again analyzed by UV–VIS and HPLC. The resulting compositions are summarized in Appendix A. The effect of SoC and temperature on AQDS stability is graphically presented also in Figure 4, where we can clearly see that AQDS concentration in the electrolyte decreases with increased SoC of electrolyte, particularly at SoC above 50% SoC, showing approx. 0.05 mol dm^−3^ decrease after five days in 100% SoC at 20 °C, which corresponds to 6% loss of theoretical initial capacity. These values correspond well with capacity decay of 4.4% observed for the identical conditions in FEC (shown in the previous section). The capacity decay was significantly accelerated at 60 °C, especially above 50% SoC. Over 23% of the initial AQDS was decomposed at 100% SoC.

In accordance to the previous results, no new compounds have been identified by HPLC analysis and the molar ratio between individual constituents remained similar for all degradation conditions. The areal % ratio of 2,6-AQDS to 2,7-AQDS increased only slightly with SoC from 0.9 (for 0% SoC samples) to 0.95 (for 100% SoC samples) indicating slightly better stability for 2,6-AQDS isomer. 2,6-AQDS and 2,7-AQDS together initially accounted for approx. 83% of all AQ derivative signals and their content slightly increased after storing for five days at high SoC to 90%. The concentration of sulfuric acid evaluated by acido-basic titration did not follow monotonous trend and changes in concentration are rather minor. The increased sulfuric acid concentration might indicate desulfonation of AQDS leading to reduced solubility and precipitation of desulfonation products (AQS or AQ). However, we did not observe solid precipitate on the bottom of the glass vial and increased content of these compounds was not observed by HPLC. Thus, the desulfonation of AQDS is not likely to significantly contribute to the degradation of electrolyte capacity. Interestingly, the content of monosulfonated 2-AQS, which is present in initial sulfonation mixture at a significant amount, decreased with increasing electrolyte SoC which is graphically shown in Figure 4. The capacity decay is thus at least partially related to decreased content of 2-AQS and thus further attention should be paid to decrease its content in the sulfonation mixture.

From the applicational point of view, our results show that acidic AQDS negolyte can be used directly from sulfonation without costly separation of the 2,7-AQDS isomer and its lifetime in RFB can be prolonged when battery operation at high SoC and temperatures is excluded. The capacity decay is clearly attributed to the chemical instability of the reduced form and electrochemical reactions has negligible effect. Thus, the battery with AQDS acidic electrolyte should be rather employed for the applications where high power densities are required but the battery is not operated in a full state of charge region.

### 2.4. DHC Parameters Evolution

The presence of DHC in the system enabled us to monitor the DHC parameters within the entire experiment to see the effect of electrolyte changes on DHC polarization. The EIS and load curves of DHC measured at 50% SoC after each series of FEC cycling together with evolution of the DHC parameters are presented in Figure 5. The SoC of 50% was adjusted by galvanostatic charging of electrolyte in FEC for the required charge according to the actual *Q*_dis._ From Nyquist spectra in Figure 5a we can see that faradaic losses dominate on the overall DHC polarization, *R*_CT_ accounting for 0.8 Ohm cm^2^ while *R*_ohmic_ only for 0.22 Ohm cm^2^. The initial overall resistance evaluated from the linear part of the load curve was approx. 1.1 Ohm cm^2^. It is worth to note that the initial *R*_CT_ is significantly higher when compared to our std. vanadium RFB full-cell where we have observed *R*_CT_ of 0.1 Ohm cm^2^ with the same electrodes and comparable concentration of active compound (1.6 mol dm^−3^ of vanadium ion for 1 e^–^ process) [22]. This is slightly surprising as according to the literature AQDS is supposed to exhibit much faster electron transfer kinetics when compared to both VRFB reactions [5]. The higher activation polarization of AQDS-based DHC can be related to the mutual interactions of AQDS molecules in concentrated solution (initially concentration of 0.86 mol dm^−3^). The dimerization of AQDS has been reported for even lower concentration of 50 mmol dm^−3^ [16]. Existence of quinhydrone, dimer of reduced and oxidized AQDS has been previously reported with equilibrium constant of 80 dm^3^ mol^−1^, exhibiting maximal concentration at 50% SoC, accounting for over 40% of total AQDS concentration [14]. In fact, we dare to hypothesize that the existence of quinhydrone can be the reason for low capacity fades of AQDS anolyte at SoC of 50% and lower, preventing the disproportionation of anthrahydroquinone and, thus, the formation of inactive anthrone. However, further investigation is required to prove this hypothesis, which is beyond the scope of our manuscript.

Within the experiment, we observed gradual increase of *R*_CT_ with time, most probably related mainly to the gradual decrease of electrolyte available capacity due to AQDS degradation. Increase of *R*_CT_ can be also caused by the deactivation of graphite felt electrodes due to change in surface functionalization of the fibers, which has been reported for vanadium RFB [23,24,25], and/or adsorption of active compounds on the electrode surface. These phenomena could be distinguished via subsequent cell characterization in fresh electrolyte, but this experiment was not performed due to the limited amount of the electrolyte available for the experiments. In contrast, *R*_ohmic_ remained unaffected which indicates good chemical stability of PFSA-based membrane in the AQDS-based electrolyte and absence of poisoning of the membrane by the electrolytes.

The EIS spectra of DHC at various electrolyte SoC and operating temperature are shown in Figure 5d. According to the expectation, the lowest *R*_CT_ was observed at 50% SC due to presence of both oxidized and reduced states in equimolar ratio. The effect SoC on *R*_ohmic_ was negligible. Increasing temperature from std. 20 °C to 40 °C resulted in substantial decrease of both ohmic and charge transfer polarization of DHC due to increased reaction kinetics and improved conductivity of the electrolyte and membrane, although the permeation of AQDS through the membrane is also increased as described in the previous section.

## 3. Materials and Methods

### 3.1. Preparation and Analysis of AQDS-Based Electrolytes

AQDS electrolyte in sulfuric acid-based supporting electrolyte was prepared by sulfonation of AQ in oleum by the following procedure: in a 1.5 l flask 202 g of anthraquinone (Sigma Aldrich, 98 wt% purity) was mixed with 1g of boric acid (99.5% purity, Lachema) which is acting as a catalyst of sulfonation and with 276 g of SO_3_ in form of oleum (58% content of SO_3_, Fluka). The mixture was mixed for 16 h at temperature of 120 °C. The fumes of SO_3_ disappeared. Subsequently, the temperature was decreased to 100 °C and 700 mL of demineralized water was poured into the flask without exothermic reaction. The mixture was mixed for 5 h until the complete dissolution was reached. Subsequently, the mixture was filtered through a dense frit (sinter S3) and water was added to the filtrate to reach a total volume of 1 dm^3^. The solid phase which remained on the frit corresponded to 2% of the initial weight of the anthraquinone, reaching the total concentration of sulfonated AQ of 1.0 mol dm^−3^. The extinction coefficient of the AQDS electrolyte based on sulfonation mixture for UV–VIS spectrophotometric analysis was 54,644.8 dm^3^ mol^−1^ cm^−1^ at wavelength of 260 nm. The concentration of residual sulfuric acid was evaluated as 1.2 mol dm^−3^ by potentiometric titration using 0.1 mol dm^–3^ NaOH. For further electrochemical experiments the electrolyte was diluted by demineralized water to 0.86 mol dm^–3^ of AQDS and 1.0 mol dm^−3^ of sulfuric acid.

For the spectrophotometric analysis the electrolyte sample was diluted approx. 50,000 times to achieve absorbance values around 1 which provides the highest precision of spectrophotometric measurement. The precise diluting procedure was performed containing 3 following steps: From the electrolyte sample of known volume and weight (i.e., known density), we weighted corresponding amount of electrolyte (with precision of four decimal points) and diluted it in a 100 mL volumetric flask by demineralised water. Subsequently, we precisely took 1.0 mL of the diluted solution to a 100 mL volumetric flask and again we diluted it by water. Finally, 5.0 mL of this diluted solution was taken to a 50 mL volumetric flask and it was diluted by water. The solution after the dilution was spectrophotometrically analyzed in two cuvettes against one blank cuvette. The difference between the measurements in the two cuvettes was not higher than ±0.003. By this approach we evaluated the content of AQDS in the analyzed electrolyte per weight and per volume. The AQDS was always analyzed in the oxidized form, even when testing the samples stored at various SoC. The full oxidation of AQDS was achieved by the oxygen dissolved in the water used for the diluting and it was proven by the repeated spectrophotometric measurement after 1 day in the presence of air.

The composition of the sulfonated AQ mixture and electrolyte samples after stability tests was evaluated by high-performance liquid chromatography (HPLC) using LC-20AD chromatograph PDA with Kinetex FS 2.6 μm 100A column (Shimadzu). Electrolyte was diluted by methanol, ammonium acetate aqueous solution (1 g dm^−3^) acted as a mobile phase A, methanol acted as mobile phase B. Detection of the constituents was performed at the wavelength of 260 nm. Relative area of the peaks corresponds to molar percent of individual sulfonated anthraquinones. HPLC chromatogram of the AQ sulfonation mixture is presented in Appendix A.

The AQ sulfonation mixture contained mainly 2,6-AQDS and 2,7-AQDS isomers and small amount of several other mono- and disulfonated derivatives of AQ, the exact composition of the mixture before and after electrochemical tests is summarized in Appendix A in the result section. In addition to these, there was a minor amount of three unknown compounds, which might be potentially three isomers of tri-sulfonated AQ, however we were not able to achieve the corresponding standards to verify this hypothesis.

Alternatively, the AQDS acidic negolyte was prepared by ion-exchange of commercially available 2,7-AQDS disodium salt (97 wt% purity, TCI) to the hydrogen form using Amberlite IR120 ion-exchange resin (Serva Feinbiochemica) and its subsequent dissolution in 1.0 M sulfuric acid. The procedure of ion-exchange was as follows: aqueous solution of 2,7-AQDS sodium salt was repeatedly flushed through the glass column containing excessive amount of ion-exchange resin. Between these cycles the column was flushed through by demineralized water (to remove residual AQDS) and subsequently by an excessive amount of 1 mol dm^−3^ sulfuric acid (to regenerate protonic form of the resin) and again by water (to remove residual traces of acid before next ion-exchange step). The ion-exchange was terminated when pH of AQDS solution reached steady value. Finally, the AQDS solution was dried in vacuum oven at 80 °C and reduced pressure and used for the preparation of electrolyte. For the preparation of electrolytes we further used sulfuric acid (96 wt%, Penta), hydrobromic acid (46–48 wt%, Verkon), and demineralized water.

### 3.2. AQDS-Br RFB Single-Cell Tests

The performance of both AQDS negolytes was compared in the AQDS/Br RFB single-cell using std. RFB lab-cell (Pinflow Energy Storage, s.r.o.). The cell consisted of copper current collectors, PPG86 composite plates (Eisenhuth), PVC-based distribution frames, polyacrylonitrile-based graphite felt electrodes (thermally activated according the procedure described in [25]). The felts were mutually separated by cation-exchange membrane F930-RFD (30 µm thick, Fumatech) with an active geometric area of 4 cm^2^. Negative electrolyte comprised of 10 mL of AQDS-based electrolyte and positive electrolyte contained 20 mL of 3.0 M HBr dissolved in deionized water. Experimental procedure consisted of EIS and load curve characterization in 50% SoC of negolyte. EIS was measured under OCV conditions with 5 mV amplitude in the frequency range of 10 kHz–20 mHz. Load curve was measured using current scan from 0 to 500 mA cm^−2^ with 5 mA cm^−2^ s^−1^ scan rate. The following parameters were evaluated from EIS and load curve measurements at various SoC: Ohmic resistance (*R*_Ω_) and charge transfer resistance (*R*_CT_) were evaluated from EIS spectra by fitting to the corresponding equivalent circuit (shown in Appendix A) and related to the geometric active area. Discharging resistance (*R*_dis_) was evaluated as a slope of discharging load curve in linear part region (0–200 mA cm^−2^) and related to the geometric active area. Finally, series of 25 galvanostatic charge-discharge cycles at 125 mA cm^−2^ was performed. Coulombic, voltage and energy efficiencies (*CE*, *UE*, *EE*) of the battery were calculated using standard equations from the charge-discharge cycles and averaged over 11th–25th cycle. Relative capacity decay per cycle (*CD*) was evaluated as a slope of dependency of discharge capacity (*Q*_dis_) on cycle number for 11th–25th cycle and related to the theoretical capacity (*Q_t_*_heor_.) according to Faraday’s law. For all experiments an electrolyte flow rate of 40 mL min^−1^ corresponding to a linear flow velocity of 54 cm min^−1^ through felt electrodes using a Watson Marlow peristaltic pump. Both electrolytes were initially purged with nitrogen (for 1 h) and nitrogen atmosphere was used within the whole experiment to prevent oxidation of reduced AQDS by atmospheric oxygen. The testing was carried out at room temperature.

### 3.3. Evaluation of Electrochemical Stability of Sulfonation Mixture

For the evaluation of electrochemical stability of AQDS-based electrolytes, we have developed an apparatus combining a flow electrolysis cell (FEC) and double half-cell (DHC), both provided by Pinflow Energy Storage, s.r.o. The apparatus is schematically depicted in Figure 2. FEC comprised of graphite felt working electrode (PAN-based, thermally activated) and platinized titanium mesh counter electrode, both mutually separated by cation-exchange membrane F1850 (50 µm thick, Fumatech) with an active geometric area of 20 cm^2^. Mercurosulphate reference electrode (sat. K_2_SO_4_ internal electrolyte with potential of 0.65 V vs. SHE) was positioned on the outlet of working electrode half-cell via Lugin capillary plugged-in through the T-junction. DHC had the identical construction as RFB cell described in 4.2. The working AQDS-based electrolyte was circulated through both half-cells of DHC and through working half-cell of FEC. Counter electrolyte consisting of 200 mL of 2.76 mol dm^−3^ sulfuric acid (without presence of electroactive compound) was circulated through counter half-cell of the electrolysis cell where hydrogen or oxygen evolution proceeded during the FEC operation. Water consumed by water splitting reactions was periodically replenished to the electrolyte tanks. Both electrolytes were circulated at 40 mL min^−1^ flow rate corresponding to linear flow velocity of 27 cm min^−1^ through the felt electrode. The apparatuses were placed in a climate box tempered at required temperature (mostly 20 °C, sometimes 40 °C). All electrochemical measurements were performed with potentiostat/galvanostat VSP300 (Biologic).

In our study, FEC was used to evaluate the available capacity and to adjust SoC of the working electrolyte while DHC was used to apply current load under constant SoC conditions and to see the impact of operating conditions on battery performance. The experimental procedure consisted of the following steps:(a)Combined galvanostatic-potentiostatic (U-I) cycling of FEC: 10 reduction-oxidation cycles were performed in full SoC range of working electrolyte to assess the actual available capacity of the working electrolyte. Each cycle consisted of galvanostatic reduction/oxidation at 150 mA cm^–2^ until corresponding potential limit was reached (reduction limit of –0.1 V and oxidation limit of +0.65 V vs. SHE) with subsequent potentiostatic hold at the limiting potential until absolute current density decrease below 5 mA cm^−2^.(b)Hold for five days at required SoC of electrolyte (0, 50, or 100%) under either OCV conditions or current load of 150 mA cm^−2^ realized by DHC battery. Electrolyte was circulated through both cells during the hold.(c)UV–VIS characterization of counter electrolyte before and after the hold was done to assess capacity losses due to permeation of AQDS through the membrane in electrolysis cell.(d)EIS and LC characterization of FEC and DHC in 50% SoC of AQDS electrolyte was done to monitor the performance evolution of both cells during the experiment. EIS was measured under OCV conditions with 5 mV amplitude in frequency range of 10 kHz–20 mHz. Load curve was measured using current scan from 0 to 500 mA cm^−2^ with 5 mA cm^−2^ s^−1^ scan rate.

The procedures a-d were repeated several times at required conditions of hold (SoC, current load, temperature).

For FEC the following parameters were evaluated from each individual series of reduction-oxidation cycling: Discharge capacity (*Q*_dis_) was evaluated as the maximal capacity of oxidation (i.e., discharge) part of the cycle. Capacity utilization (*CU*) was evaluated as the ratio of *Q*_dis_ vs. initial *Q*_theor_. *CE*, *UE*, *EE* of the working half-cell of FEC were calculated from working electrode potential evolution during reduction-oxidation cycles and averaged over 6th–10th cycle of each series. Relative capacity decay per cycle (*CD*) was evaluated as slope of *Q*_dis_-cycle number dependency for 6th–10th cycle and related to *Q_t_*_heor_. The capacity fade after the hold at given conditions Δ*Q*_cap._ was calculated by subtracting *Q*_dis_ of cycling series after and before the hold. The capacity loss related to permeation of AQDS through the membrane of FEC Δ*Q_perm._* was calculated from difference in UV–VIS absorbance of counter electrolyte before and after the hold using the calibration curve using UV–VIS spectrometer (Agilent 8453) in a quartz glass cuvette with the optical length 1.0 cm. The calibration curve was measured using the initial electrolyte diluted to required concentrations in the range of 0.1–0.5 mM of AQDS in the 2.76 mol dm^−3^ H_2_SO_4_ supporting electrolyte. The concentration of AQDS was evaluated from local absorbance maximum at 258 nm. All the relative capacity fades are related to *Q*_theor_.

The DHC resistances were evaluated from EIS and lad curved measurements by the same procedure as for AQDS-Br RF, described in Section 2.3.

## 4. Conclusions

We studied the electrochemical stability of acidic AQDS-based negolyte directly prepared from anthraquinone sulfonation mixture for the application in acidic RFB with HBr based catholyte. We found that sulfonation mixture of AQ containing mainly 2,6-AQDS and 2,7-AQDS isomers can be directly used as AQDS acidic negolyte providing high capacity utilization (95%) and comparable cell parameters in an AQDS—HBr RFB single-cell when compared to the corresponding negolyte based on pure 2,7-AQDS. The electrochemical stability of the negolyte based on the sulfonation mixture was further investigated under various operating conditions. For that purpose, a novel method using combination of flow electrolysis cell and a double half-cell was designed and used enabling differentiation of individual contributions to capacity fade including chemical and electrochemical degradation and cross-over of active species through the membrane. The effect of electrolyte SoC, operating temperature and current load applied on the double half-cell was systematically studied. The results showed that the purely chemical degradation of AQDS takes place at SoC above 50% and is significantly enhanced at elevated temperature. The post mortem HPLC analysis of electrolytes revealed deep degradation of electrolyte active compounds leading to loss of aromaticity of the degradation products. The capacity decay is also, at least partially, caused by decreased content of anthraquinone-2-sulfonic acid, which is contained in the initial sulfonation mixture at a significant amount. Good chemical stability at 50% SoC with negligible capacity decay of 0.02% per day might be associated to the presence of quinhydrone (complex of oxidized and reduced form) preventing the degradation of anthrahydroquinone. The developed method can be adopted for a detailed investigation of capacity fades for various RFB chemistries.

## Figures and Tables

**Figure 1 molecules-26-02484-f001:**
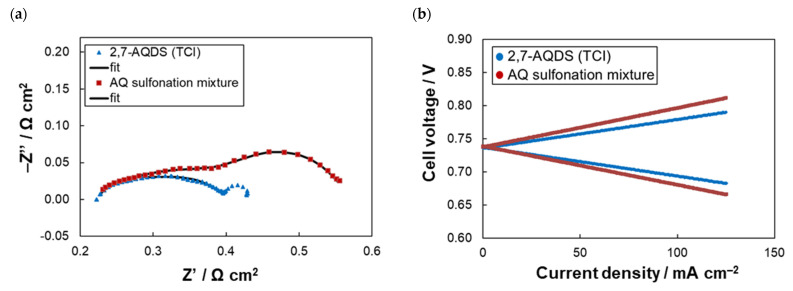
Performance of AQDS-HBr single-cell using two types of anolyte: (**a**) EIS Nyquist spectra of a single-cell in 50% SOC, (**b**) Load curve of single-cell in 50% SoC, (**c**) U-Q dependence of galvanostatic charge-discharge cycle (3rd cycles) at 125 mA cm^−2^; (**d**) Evolution of battery parameters during the cycling.

**Figure 2 molecules-26-02484-f002:**
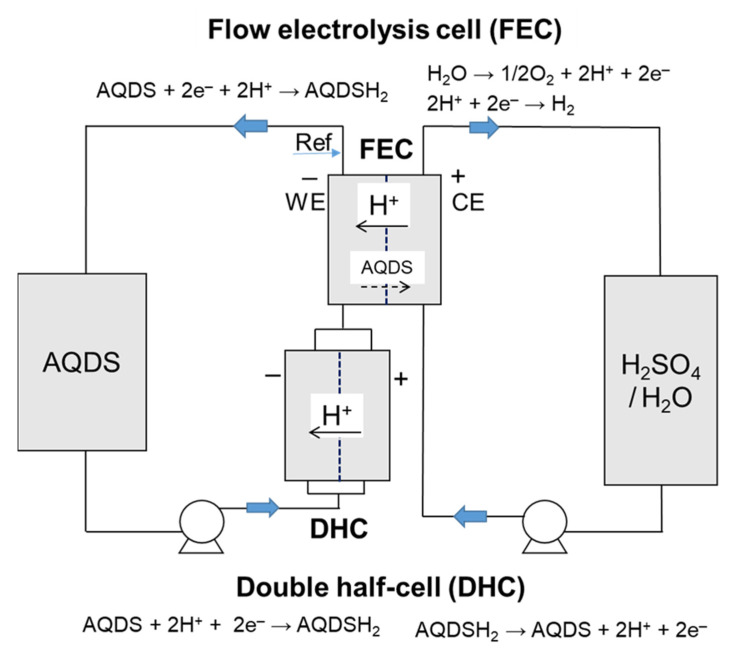
Scheme of experimental set-up for evaluation of the electrochemical stability of AQDS electrolyte.

**Figure 3 molecules-26-02484-f003:**
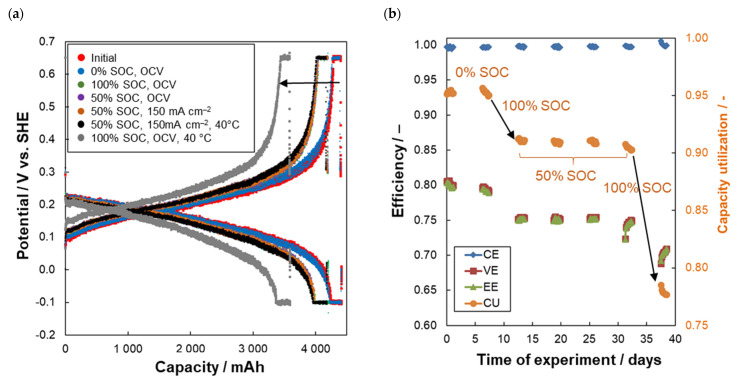
(**a**) Reduction-oxidation curves (6th cycle) from FEC at 150 mA cm^−2^ after five day pause at the specified conditions (see legend); (**b**) Evolution of parameters of electrolysis cell during series of cycling during the whole experiment.

**Figure 4 molecules-26-02484-f004:**
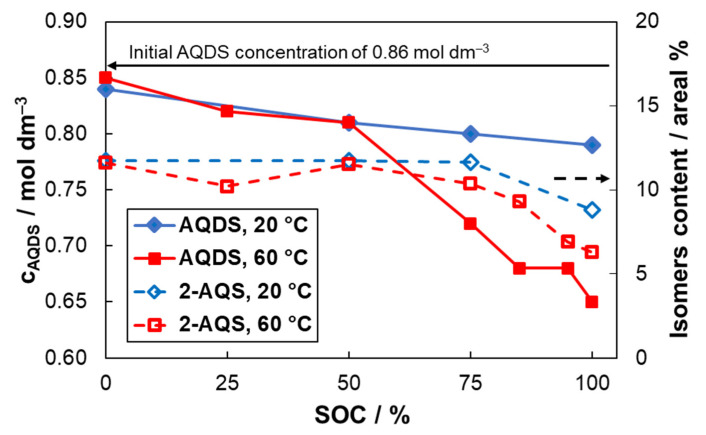
The effect of electrolyte SoC and storing temperature on the concentration of AQDS from UV–VIS spectrophotometry (primary *y*-axis) and areal % of 2-AQS (secondary *y*-axis) from HPLC after five days in a glass vial (initial SoC adjusted by charging in std. AQDS-HBr RFB single-cell).

**Figure 5 molecules-26-02484-f005:**
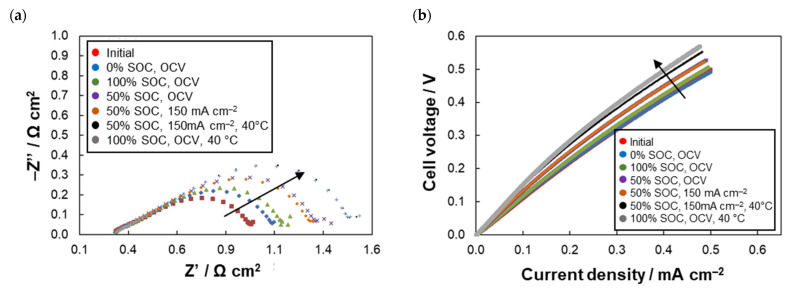
EIS spectra (**a**) and load curves (**b**) of DHC in 50% SoC after individual holds and development of evaluated resistances during experiment (**c**); EIS Nyquist spectra in different electrolyte SoC and temperatures (**d**).

**Table 1 molecules-26-02484-t001:** AQDS-HBr single-cell parameters with different AQDS-based anolytes.

Anolyte	*R*_Ω_^(1)^ (Ω cm^2^)	*R*_CT_^(1)^ (Ω cm^2^)	*R*_dis_^(2)^ (Ω cm^2^)	*CE*^(3)^ (%)	*VE*^(3)^ (%)	*EE*^(3)^ (%)	*CU*^(3)^ (%)	*CD*^(3)^ (%*Q_t_*_heor_/Cycle)
2,7-AQDS (TCI)	0.22	0.18	0.44	98.6	84.3	83.2	97.0	–0.45
AQ sulfonation mixture	0.20	0.38	0.58	98.8	78.1	77.2	96.1	–0.34

Evaluated from EIS^(1)^, discharge load curve^(2)^ at 50% SoC and galvanostatic cycling^(3)^ at current density of 125 mA cm^−2^.

**Table 2 molecules-26-02484-t002:** Averaged parameters of flow electrolysis working half-cell within whole experiment.

Series nr.	*CE* (%)	*VE* (%)	*EE* (%)	*CU* (%)	*CD* (%*Q*_theor_/Cycle)	Δ*Q*_cap._ (%*Q*_theor_/hold)	Δ*Q*_perm._ (%*Q*_theor_/hold)	Conditions:
1	99.7	80.0	79.8	95.5	–0.08	–	–	Initial
2	99.7	79.3	79.0	95.7	–0.08	0.17	0.004	After five days at 0% SoC, OCV
3	99.7	75.4	75.2	91.3	0.01	–4.38	0.002	After five days at 100% SoC, OCV
4	99.7	75.2	75.0	91.1	0.00	–0.21	0.002	After five days at 50% SoC, OCV
5	99.7	75.4	75.2	91.2	0.02	0.10	0.002	After five days at 50% SoC, 150 mA cm^–2^
6	99.7	74.9	74.7	90.8	–0.03	–0.38	0.020	After five days at 50% SoC, 150 mA cm^–2^, 40 °C
7	99.9	70.7	70.6	78.5	–0.10	–12.3	0.023	After five days at 100% SoC, OCV, 40 °C

## Data Availability

All the data supporting reported results can be found in Appendix A.

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
