# Peer review of "Evaluation of Electrochemical Stability of Sulfonated Anthraquinone-Based Acidic Electrolyte for Redox Flow Battery Application"

_molecules, 2021, doi:10.3390/molecules26092484_

Round 1
Reviewer 1 Report
This manuscript describes a method to evaluate the chemical stability of acidic electrolyte for aqueous organic redox flow batteries. By using the UV-vis spectroscopy and HPLC, this work testify their methods in studying the stability of AQDS-based acidic electrolyte. However, in its conclusion, the new method is not compared with existing methods for evaluating electro-chemical stability. AQDS-based acidic electrolyte. I would recommend the authors to revise the manuscript significantly. Some technical comments:
1) Figure.4, the caption is unclear “initial electrolyte and electrolyte after electrochemical test, diluted 50 000 times by demineralized water, measured in cuvette with the optical length of 1 cm.” The comparison for absorbance intensity is not convincing by showing only two curves without mentioning the uncertainties in measurements. It is not clear if the difference is due to the dilution or the electro-chemical reactions.
2) On page 9, line 261 “Nor..” is unclear to me. HPLC analysis identified or did not identify new compounds?
4) The manuscript should be shortened to show the major results and highlight the major findings that are more advantageous over conventional methods.
Reviewer 2 Report
This study reported the evaluation of electrochemical stability of sulfonated anthraquinone-based acidic electrolyte for redox flow battery application. The following comments should be considerably addressed before accepting for publication.
- The authors listed the acronyms in Appendix A. However, it would be better to explain in the context for the first time.
- The equivalent circuit used for fitting the EIS spectra (Fig. 1a, 6a, 6d) should be presented for better understanding.
- In conclusions, the results showed that the purely chemical degradation of AQDS takes place at SoC above 50% and is significantly enhanced by elevated temperature. It is better to discuss or comment on how to increase the chemical stability or SoC for such an electrolyte system.
- The supplementary materials were not available for the review procedure.
Round 2
Reviewer 1 Report
All the questions have been addressed. I think now it can be accepted for publication.
Reviewer 2 Report
The comments were addressed accordingly. It would be accepted for publication now.